# Plant Pellets: A Compatible Vegan Feedstock for Preparation of Plant-Based Culture Media and Production of Value-Added Biomass of Rhizobia

**Hassan-Sibroe A. Daanaa** [1,*,†], **Mennatullah Abdou** [1,‡], **Hanan A. Goda** [1],
**Mohamed T. Abbas** [2], **Mervat A. Hamza** [1], **Mohamed S. Sarhan** [1,§], **Hanan H. Youssef** [1],
**Reem Hamed** [1], **Mahmoud El-Tahan** [3], **Mohamed Fayez** [1], **Silke Ruppel** [4] **and Nabil A. Hegazi** [1]

1   Department of Microbiology, Faculty of Agriculture, Cairo University, Giza 12613, Egypt;
    menna.abdou88@yahoo.com (M.A.); Hanan.goda@agr.cu.edu.eg (H.A.G.);
    mervathamza66@gmail.com (M.A.H.); m.sabrysarhan@gmail.com (M.S.S.);
    hananyoussef16@gmail.com (H.H.Y.); reemhamed1812@gmail.com (R.H.); mfayezgiza@yahoo.co.uk (M.F.);
    hegazinabil8@gmail.com (N.A.H.)
2   Department of Microbiology, Faculty of Agriculture & Natural Resources, Aswan University,
    Aswan 81528, Egypt; mtabbas67@gmail.com
3   Regional Center for Food & Feed (RCFF), Agricultural Research Center, Giza 12619, Egypt;
    eltahanmh@gmail.com
4   Leibniz Institute of Vegetable and Ornamental Crops (IGZ), 14979 Grossbeeren, Germany; ruppel@igzev.de
*   Correspondence: hsdaanaa@nig.ac.jp; Tel.: +81-070-4215-5672
†   Current Address: Department of Genetics, School of Life Science, SOKENDAI (The Graduate University for
    Advanced Studies), Mishima 411-8540, Japan.
‡   Current Address: Department of Biochemistry, Faculty of Agriculture, Cairo University, Giza 12613, Egypt.
§   Current Address: Eurac Research, Institute for Mummy Studies, 39100 Bolzano, Italy.

**Abstract:** Although plant-based culture media enhances in vitro cultivation of rhizobacteria, studies assessing their biomass potential for large-scale applications are lacking. Here, we advance plant pellets (PPs) as a novel technology to unlock the potential of such vegan culture media for biomass production of *Rhizobium leguminosarum*. PP formulations were based on mixtures of Egyptian clover powder and the agro-byproducts glycerol and molasses. These mixtures were either contained or not contained in teabags during culture media preparation. Metrics of biomass included colony forming units, optical density ($OD_{600nm}$), and cell dry weight (DW). Biomass comparisons between culture media based on PPs and standard yeast extract mannitol (YEM) revealed that the following PPs composition, contained in teabags, cultivated rhizobia at levels comparable to YEM: 16 g clover powder, 5% molasses, and 0.8% glycerol. This PPs composition enabled shorter generation times of rhizobia (PP: 3.83 h, YEM: 4.28 h). Strikingly, PPs mixtures supplemented with 10% molasses and not contained in teabags promoted rhizobia without apparent lag phases and produced 25% greater DW than YEM. PPs potentiate the use of dehydrated vegan feedstocks for both plant microbiota cultivation and biomass production and appear as cost- and labor-effective tools, easy to handle and store for plant-based culture media preparation.

**Keywords:** biomass production; plant-based culture media; plant pellets; *Rhizobium leguminosarum*; sustainable agriculture; feedstocks for rhizobacteria

## 1. Introduction

The plant microbiome plays a pivotal role in enhancing plant health and nutrition. Here, the plant root exudates, rich in organic acids and proteins, enable mutually beneficial interactions between

microbial consortia and their host plants [1–6]. Reciprocally, phytohormone production, nitrogen fixation, pathogen antagonism, and facilitation of nutrient availability and acquisition are among the multiplicity of microbiome functions that increase plant productivity. Hence, core microbiomes are crucial bioresources for sustainable agriculture and related strategies [7–10]. Particular attention has been given to beneficial members of the plant microbiome stationed in the root zone (rhizosphere), with particular emphasis on plant-growth-promoting rhizobacteria (PGPRs). Common classifications of PGPRs are bio-fertilizers, that fix or solubilize nutrients for plant consumption, and bio-pesticides, which function to eliminate phytopathogens and enhance plant resistance [11]. A prominent example of PGPRs are members of *Rhizobium* spp. that fix atmospheric nitrogen as part of a symbiotic process involving penetration into root cells and formation of root nodules [12,13]. Other members of *Pseudomonas* spp., *Azospirillum* spp., *Azotobacter* spp., and *Enterobacter* spp. are also involved in nitrogen fixation through associated and/or asymbiotic interactions [14]. Members of these genera (including rhizobia) also include phytostimulatory agents that secrete—e.g., indole acetic acid and ACC deaminase—and/or mobilize elements such as P, K, and Fe absorbed by plants [14,15]. Given these features of agro-biopreparates (bio-fertilizers and bio-pesticides), their large-scale cultivation and application has become a central agro-biotechnological tool supporting the productivity of economically important crops, especially staple food sources [14,16]. Thus, PGPRs are vital bioresources towards the end goal of satisfying global food demand [17,18].

Developing culture media that can cultivate PGPRs is a fundamental prerequisite for large-scale production of agro-biopreparates. Although culture media based on pure chemicals support the development of PGPRs, a high cost of production renders them less favorable to use for industrial settings, especially in developing countries. In this respect, acquiring cheap substrates with rich nutritional profiles is a critical step towards the cost-effective production of sufficient quantities of agro-biopreparates [19–21]. In particular, the byproducts of agro-industries (agro-byproducts) satisfy these characteristics, being cheap, easily obtainable, and possessing multiple carbon and nitrogen sources in organic forms as well as vitamins and minerals necessary for microbial growth [21–25]. Notable examples of the application of agro-byproducts include the use of effluents of the bakers' yeast industry to grow a diverse array of bio-fertilizers of *Azotobacter chroococcum* and *Rhizobium leguminosarum* [22]. In addition, spent wash, molasses, and glycerol have been demonstrated to increase the growth rate, biomass yield, and nodulation efficiency of both fast- and slow-growing rhizobia, and also to produce biosurfactants of *Bacillus* spp. [26–29].

Plant materials/residues have also emerged as economic and efficient substrates for the production of rhizobacteria. Culture media preparations based on juices of desert plants *Mesembryanthemum crystallinum* L., *Zygophyllum album* L., and *Carpobrotus edulis* L. contained ample nutrient stores that supported the good growth of facultative nitrogen fixers such as *Enterobacter agglomerans* and *Azospirillum brasilense* [30]. The extracts of onions, tomatoes, and beans have all been used to study the growth of *Pseudomonas* spp. and its biocontrol properties [31]. In particular, our recent work demonstrated that culture media prepared from the juices or slurries of *Trifolium alexandrinum*, *Paspalum vaginatum*, and *Opuntia ficus-indica* support in vitro growth and biomass production of a number of model rhizobacteria [32]. In addition, innovations to facilitate the application of plant-only-based culture in the form of plant powder teabags resulted in improving in vitro cultivability of rhizobacteria and also allowed investigations on their diversity and, to a lesser extent, the biomass potential [33–36]. Taken together, utilization of plant materials for culture media preparation offers a promising and relatively inexpensive strategy towards biomass production of agro-biopreparates.

While plant materials in the forms of juices, slurries, and powders in teabags have been used for the application of plant-only-based culture media to improve in vitro cultivability of rhizobacteria [33–36], no studies have (1) directly assessed the potential of such media to produce biomass and (2) integrated the rich nutrient stores of plant materials with agro-byproducts to produce value-added biomass.

Efforts to establish plant materials as major substrates for the large-scale production of PGPRs would benefit from assessing biomass production on plant-based culture media.

To assess and facilitate the application of plant-based culture media for biomass production of PGPRs for basic and applied research, we developed a novel technology called plant pellets (PPs). PPs represent a form of dehydrated plant material that can be used on their own or in combination with agro-byproducts. Since PPs can be stored, they shorten the process of preparing plant-based culture media, thus, encouraging the use of PPs in industries, where the preparation time of culture media can be a major limitation. This study presents PPs processed from various combinations of dehydrated powders of Egyptian clover, *Trifolium alexandrinum*, and supplements of nutritionally rich local agro-byproducts, molasses and glycerol. Formulated PPs were either contained or not contained in teabags during culture media preparation. For assessments, PPs culture media were compared to the standard yeast extract mannitol (YEM) in supporting cell growth and biomass production of the model plant: symbiotic rhizobia (*Rhizobium leguminosarum* bv. trifolii USDA 2128). For this purpose, batch cultures of PPs and YEM were prepared to monitor bacterial growth in forms of colony forming units (CFUs), optical density ($OD_{600nm}$), estimated nonviable cells (nVCs), and dry weight (DW) for up to 15 days of incubation. In addition, we measured the electric conductivity and pH and their coefficients of variation to further explore properties of PP culture media.

## 2. Materials and Methods

### 2.1. Experimental Design

In view of our previous publications [32–35,37], we set up a number of preliminary experiments to assess the growth of *Rhizobium leguminosarum* and biomass production on the expense of increasing concentrations of dehydrated clover plant powder and boosting effects of supplementation with rich and available agro-byproducts of local molasses and glycerol in various concentrations. Then, the main experiments were designed to measure the suitability of supplemented concentrations of molasses and glycerol not only to enrich the plant material but also to formulate stable and consistent plant pellets. The produced plant pellets, contained in teabags, were further used to prepare plant-based culture media and to support growth and biomass production of the tested rhizobia isolate compared to the standard YEM culture medium. Further, trials were carried out to fortify the nutritional content of plant pellets by increasing percentages of added molasses and comparing the effect of direct inclusion of PPs to those contained within teabags as well as standard YEM culture medium.

### 2.2. Rhizobia Strain

The representative isolate of *Rhizobium leguminosarum* bv. trifolii (USDA 2128) was used. This particular isolate is commercially produced, and commonly used to inoculate clover and related legume plants (e.g., alfalfa, faba bean, and pea) in many countries. The strain is also commonly used to inoculate Egyptian clover (Berseem, *Trifolium alexandrinum* L., Family Fabaceae), especially when grown in newly reclaimed desert sandy soils [38,39]. It was obtained from the culture collection of the Department of Microbiology, Institute of Soil and Environmental Sciences, Agricultural Research Center, Giza, Egypt.

### 2.3. Plant Materials

The whole vegetative parts (stems and leaves) of full-grown Egyptian clover were collected from the experimental fields of the Faculty of Agriculture, Cairo University, Giza, Egypt (30.0131° N, 31.2089° E), during the winter season (January–March). According to Sarhan et al. [33], plant samples were dehydrated in sunlight for 24–48 h and then subjected to oven drying at 70 °C for 24–48 h. The dried materials were ground before passing through a sieve with a pore size of 2 mm to obtain fine clover powder.

## 2.4. Agro-Industrial Byproducts

The crude molasses was obtained from the "Sugars and Integrated Industries Egyptian Distillation Plants" in Hawamdeyah, Giza, Egypt, and was stored at −20 °C until use. It was used as a supplement to the plant powder for pellet processing and culture media formulation. To ensure molasses sterility, it was subjected to three intermittent cycles of sterilization (at 121 °C for 20 min) over three successive days.

As a byproduct of agro-industries, glycerol was included as an additional amendment and as a gelling substance for pellet texture and consistency due to its viscous nature [28,40]. Furthermore, glycerol represents a preferred carbon source for a variety of *Rhizobium* spp. including *R. leguminosarum* [26,28].

## 2.5. Formulation of Plant Pellets (PPs) and Derived Culture Media

To prepare the tested formulated PP culture media, 120 g of the prepared clover powder was manually mixed with glycerol (0.8% (*v/v*)), molasses (5% or 10%, (*v/v*)) and ca. 250 mL of distilled water to attain sufficient moisture and texture for pellet processing. The resulting mixture was fed into a manual meat grinder to process PPs. The resulting PPs were left to dry at ambient temperature (>20–30 °C) for 48 h before use. They were >0.3–0.5 cm in diameter and >1.5–3.0 cm in length (Figures 1, 2a).

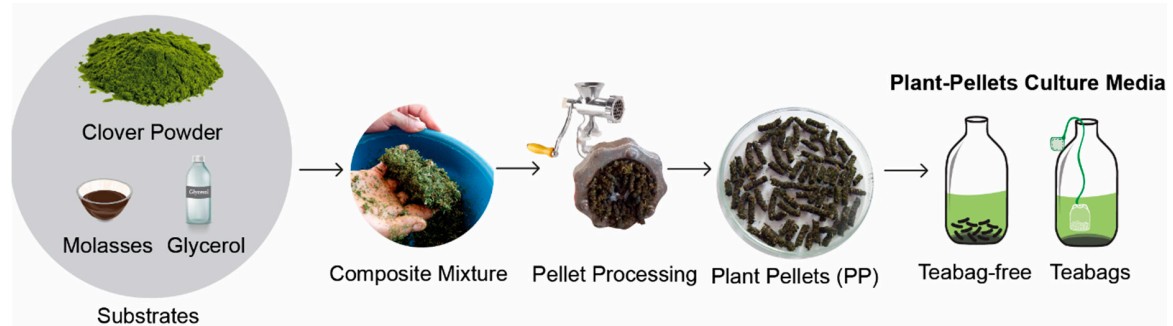

**Figure 1.** A workflow of formulation of plant pellets from mixtures of dehydrated plant powders, molasses, and glycerol; manual processing with meat grinder; and preparation of plant pellets (PP)-based culture media.

To prepare the PP liquid infusion, 8 g or 16 g of pellets were packaged in teabags and placed in Erlenmeyer flasks containing 1 L of distilled water. The infusion was heat-cooked in an autoclave (121 °C for 6 min). After cooling, the teabags were removed, and aliquots of 250 mL of liquid infusion were distributed in 500-mL Erlenmeyer flasks. Culture media formulations based on the direct inclusion of PPs into distilled water without containment in teabags (teabag-free) were also prepared.

As shown in Table 1, six different culture media combinations were prepared to determine the growth and biomass production of the tested rhizobium.

For preparation of solid culture media, agar-agar was added (15 g $L^{-1}$). All prepared culture media were adjusted to pH 7.0–7.3 before being autoclaved for 20 min at 121 °C.

For comparisons, yeast extract mannitol (YEM) was used as a recommended culture medium for rhizobia cultivation [41] and consisted of (g $L^{-1}$) mannitol, 10; $K_2HPO_4$, 0.5; $MgSO_4.7H_2O$, 0.2; NaCl, 0.1; and yeast extract, 0.5.

**Table 1.** Abbreviations of tested culture media prepared from different formulations of plant pellets (PPs) and the standard culture medium, yeast extract mannitol (YEM).

| PPs Formulations | Application Method | Culture Medium Abbreviation * |
|---|---|---|
| 8 g of plant pellets formulated with 5% molasses<br>8 g of plant pellets formulated with 10% molasses<br>16 g of plant pellets formulated with 5% molasses<br>16 g of plant pellets formulated with 10% molasses | Within teabags | $PP_{5\%m}$ 8 g L$^{-1}$<br>$PP_{10\%m}$ 8 g L$^{-1}$<br>$PP_{5\%m}$ 16 g L$^{-1}$<br>$PP_{10\%m}$ 16 g L$^{-1}$ |
| 8 g of plant pellets formulated with 10% molasses<br>16 g of plant pellets formulated with 10% molasses | Without teabags | $PP_{ntb10\%m}$ 8 g L$^{-1}$<br>$PP_{ntb10\%m}$ 16 g L$^{-1}$ |
| Yeast extract mannitol | | YEM |

* All PP culture media formulations are supplemented with 0.8% glycerol.

### 2.6. Experimental Units

Preliminary experiments were performed to evaluate the biomass production of *R. leguminosarum* throughout 7 days of incubation in plant-based culture media prepared from a mixture of different concentrations of clover plant powder (4, 8, or 16 g L$^{-1}$), with or without molasses (1, 2.5, and 5% (*v/v*)), and 0.8% glycerol. Formulations producing optimum biomass, based on dry weight, were selected for further PPs processing (Figure S1). The tested concentrations of glycerol (0.8%) and/or molasses (10% (*v/v*)) were selected based on recent studies indicating their respective optimal concentrations for microbial biomass production [29,42,43]. Results of the preliminary experiments are based on two biological replicates for each culture medium.

The first main experiment was designed to assess in vitro cultivability and biomass production of PP culture media formulations based on 8 and 16 g of pellets contained in teabags ($PP_{5\%m}$ 8 g L$^{-1}$ and $PP_{5\%m}$ 16 g L$^{-1}$, respectively) compared to standard YEM. Two biological replicates were prepared for each treatment.

The second main experiment was planned to improve the biomass productivity of the PP culture media. We assessed the efficacy of increasing the molasses concentrations in PPs up to 10% ($PP_{10\%m}$ 8 g L$^{-1}$ and $PP_{10\%m}$ 16 g L$^{-1}$). To facilitate direct access and possible immobilization onto plant materials, plant pellets were used directly, not contained in teabags, to prepare the culture media ($PP_{ntb10\%m}$ 8 g L$^{-1}$ and $PP_{ntb10\%m}$ 16 g L$^{-1}$). Both the standard YEM and tested plant pellets (within teabags), $PP_{5\%m}$ 8 g L$^{-1}$ and $PP_{5\%m}$ 16 g L$^{-1}$, were also included for comparison. Two biological replicates were prepared for each treatment.

### 2.7. Biomass Production of Rhizobium Leguminosarum on Tested Culture Media

The prepared batch cultures, 500-mL Erlenmeyer flasks containing 250 mL of culture media, were inoculated with the tested rhizobia isolate (5% (*v/v*)). Flasks were incubated at 30 °C in a rotary shaker (100 rpm) for a maximum of 15 d. The resulting batch cultures were sampled along the incubation period to measure colony forming unit counts, optical density, cell dry weight, electric conductivity, and pH.

The optical density ($OD_{600nm}$) was measured at a 600-nm wavelength using a Novaspec II visible spectrophotometer (Pharmacia LKB, Cambridge, England). $OD_{600nm}$ was measured in two technical replicates.

To measure the cell dry weight (DW), samples of 50 mL of each of the tested liquid batch cultures were centrifuged at 7000 rpm for 10 min. The resulting cell pellets were harvested and oven-dried at 70 °C for weighing. The DW was measured in three technical replicates.

To count colony forming units (CFUs), the drop plate method was used. Necessary 10-fold serial dilutions were prepared from samples representing various batch cultures. Aliquots of 60 µL from suitable dilutions were spotted onto the surface of agar plates prepared from the corresponding culture media. The plates were incubated at 30 °C for 24–96 h for CFUs counting in four technical replicates.

Growth rates and doubling times were calculated as follows [44]:

$$\text{Growth rate (K)} = \log Nt - \log No / \log 2 \qquad (1)$$

$$\text{Doubling time (dt)} = 1/K \qquad (2)$$

where No = initial viable cell counts, Nt = final viable cell counts, and t = time in hours.

To infer the proportion of nonviable cells (nVCs), standardized CFU values were used as a proxy for viable cells (VCs) and were subtracted from standardized $OD_{600nm}$, representing total cells in the same culture medium, assuming minimal interference from cell debris and heterologous proteins with the $OD_{600nm}$ measurements. Values greater than the pooled mean (0) indicate greater proportions of nVCs while values smaller than this mean indicate fewer proportions of nVCs.

The electric conductivity (EC) and pH were monitored to identify PP culture media characteristics, reflecting on-going metabolic activities. Samples of batch cultures were subjected to EC and pH measurements using a Jenway electrochemistry analyzer, Staffordshire, UK. The change and maximal change in pH ((pHfin–pHmin), (pHmin–pH0)) and EC ((ECfin–ECmin), (ECmin–EC0)) were calculated according to Colombié et al. [45], where subscript "0" is time at 0 h and "fin" denotes final time. EC and pH were each measured in two technical replicates.

## 3. Statistical Analyses

Recorded observations of $OD_{600nm}$, DW (g $L^{-1}$), and Log CFU $mL^{-1}$ were analyzed based on one-way analysis of variance (ANOVA) to discern the culture medium effect and two-way ANOVA to detect the combined effect of culture medium and incubation time on the development and biomass production of *R. leguminosarum*. Post-hoc tests for these analyses were based on Fischer LSD (least significant difference). Pearson correlations were used to determine relationships among the tested variables for biomass. Principal component analysis (PCA) was used to visualize similarities/dissimilarities between YEM and PP cultures in terms of all tested variables.

All statistical analyses were conducted using STATISTICA V10 (Statsoft Inc., Tulsa, OK, USA). The R-CRAN (cran.r-project.org) packages of "ggplot2", "stringi", "Hmisc", and "ggfortify" were used to produce boxplots, scatterplots, and PCA.

## 4. Results

### 4.1. Glycerol and Molasses Stimulate Biomass Production of Rhizobia and Facilitate Processing of Plant Pellets

Introductory experiments were carried out to assess the growth and biomass production of *R. leguminosarum* on culture media formulated from clover powder and/or clover powder supplemented with the agro-byproducts, glycerol and molasses. Among various clover powder concentrations, the abundant nutrient stores of 16 g $L^{-1}$ supported the highest biomass production, reaching ca. 5 g $L^{-1}$ (Figure S1a). Similar to previous reports, the positive value-added effects of the tested agro-byproducts of glycerol and molasses on rhizobia biomass production were observed. To combine the positive effects of both plant powder and glycerol/molasses supplements, various formulations were tested. The highest yield (>20 g $L^{-1}$) was reported for clover powder (16 g $L^{-1}$) + molasses (5% (*v/v*)), with no significant effect attributed to glycerol (0.8% glycerol) (Figure S1b). However, taking advantage of the viscosity and gelling potential of glycerol and molasses, both were further used for processing of intact plant pellets.

### 4.2. Culture Media Prepared from Plant Pellets Contained in Teabags Supported Good Growth and Biomass Production of Rhizobium Leguminosarum

Compared with standard YEM, CFU counts showed that culture media based on teabags containing PPs, developed with 5% molasses, supported normal and good growth of rhizobia. The developed cells showed the characteristic appearance of active motile short rods and when surface-inoculated

on respective agar plates, they developed normal CFU morphologies with a relatively less slimy appearance (Figure S2). Counts of CFUs developed on PP culture media and YEM were similar (ca. > Log 7.0 CFU mL$^{-1}$ culture) throughout the incubation period (Figure 2b), and no significant differences were attributed to the independent effect of culture media (one-way ANOVA, Table 2).

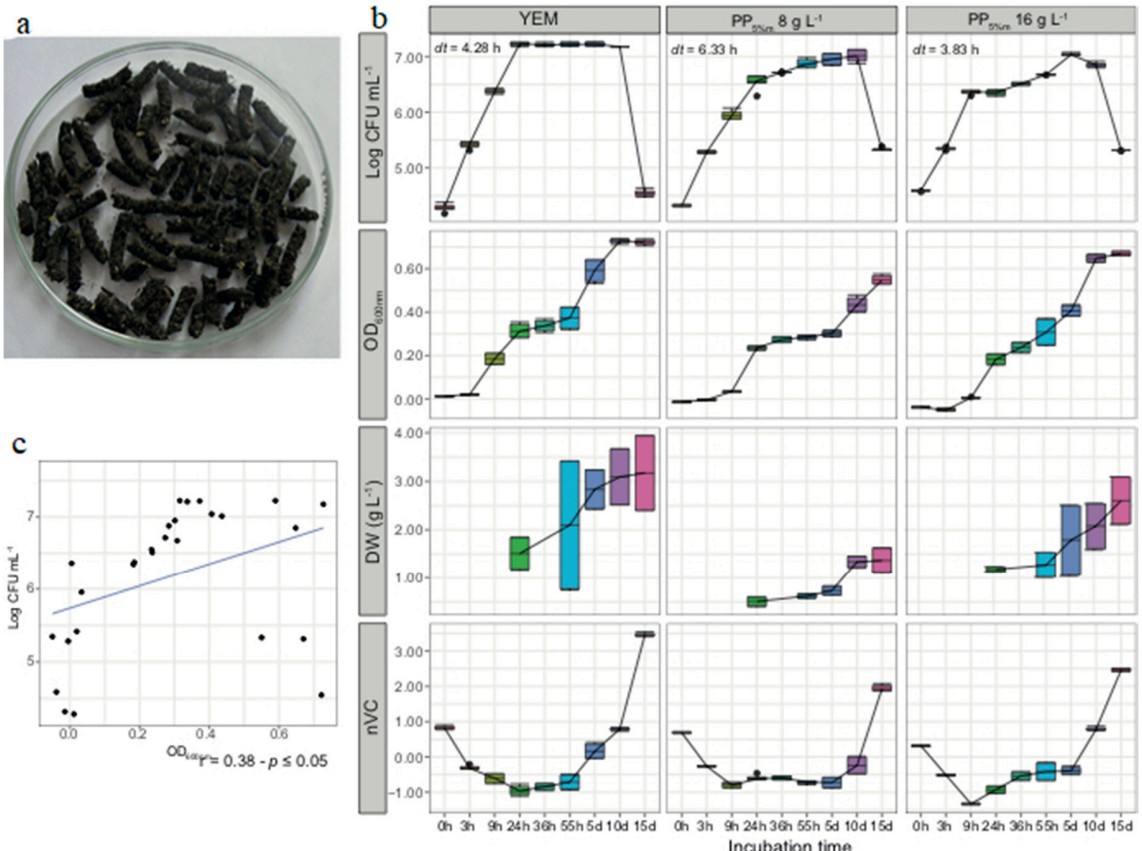

**Figure 2.** The effect of processed plant pellets (PPs) in their various formulations on growth of *R. leguminosarum* in batch cultures compared to standard YEM. (**a**) The appearance of PPs successfully pilot-produced using a manual meat grinder to process dehydrated clover plant powder mixed with crude molasses (5–10%, *v/w*) and glycerol (0.8%, *v/w*); (**b**) boxplots of growth levels of *R. leguminosarum* along the incubation period based on the four tested parameters of colony forming units (Log CFU mL$^{-1}$), optical density (OD$_{600nm}$), dry weight (DW, g L$^{-1}$) and estimates of nonviable cells (nVCs); (**c**) scatter plot of OD$_{600nm}$ and Log CFU mL$^{-1}$ values reported across the incubation period. YEM, yeast extract mannitol; PP$_{5\%m}$ 8 g L$^{-1}$, culture media based on 8 g of plant pellets formulated with 5% molasses concentration (within teabags); PP$_{5\%m}$ 16 g L$^{-1}$, culture media based on 16 g of plant pellets formulated with 5% molasses concentration (within teabags). All tested pellet formulations included 0.8% glycerol.

**Table 2.** One- and two-way ANOVA of optical density ($OD_{600nm}$), dry weight (DW), colony forming unit (CFU) counts, and estimates of nonviable cells (nVCs) of *R. leguminosarum* grown on various combinations of plant pellets (PPs) based culture media and the standard YEM. The least significant difference (LSD) is shown for each analysis.

| ANOVA: Single Effect (Culture Media) | $OD_{600nm}$ | DW (g L$^{-1}$) | Log CFU (mL$^{-1}$) | nVCs |
|---|---|---|---|---|
| YEM | 0.36 [a] | 2.54 [a] | 6.29 [a] | 0.20 [a] |
| PP$_{5\%m}$ 8 g L$^{-1}$ | 0.23 [c] | 0.91 [c] | 6.11 [a] | −0.14 [a] |
| PP$_{5\%m}$ 16 g L$^{-1}$ | 0.26 [b] | 1.78 [b] | 6.11 [a] | −0.06 [a] |

| ANOVA: Two-way Interaction (Culture Media and Time) | $OD_{600nm}$ | | | | |
|---|---|---|---|---|---|
| | Time (day) | | | | |
| | **1** | **2** | **5** | **10** | **14** |
| YEM | 0.3153 [gh] | 0.3722 [fg] | 0.5895 [cd] | 0.7272 [a] | 0.7218 [a] |
| PP$_{5\%m}$ 8 g L$^{-1}$ | 0.2345 [ij] | 0.2840 [hi] | 0.3015 [hi] | 0.4363 [e] | 0.5502 [d] |
| PP$_{5\%m}$ 16 g L$^{-1}$ | 0.1825 [j] | 0.3080 [h] | 0.4067 [ef] | 0.6480 [bc] | 0.6700 [ab] |
| **LSD (0.05)** | | | 0.06042 | | |
| | DW (g L$^{-1}$) | | | | |
| YEM | 1.510 [cde] | 2.089 [bc] | 2.832 [a] | 3.092 [a] | 3.171 [a] |
| PP$_{5\%m}$ 8 g L$^{-1}$ | 0.5047 [h] | 0.6193 [gh] | 0.7320 [fgh] | 1.337 [efg] | 1.368 [def] |
| PP$_{5\%m}$ 16 g L$^{-1}$ | 1.177 [efgh] | 1.271 [efg] | 1.777 [cde] | 2.070 [bcd] | 2.604 [ab] |
| **LSD (0.05)** | | | 0.7196 | | |
| | nVCs | | | | |
| YEM | −0.9512 [i] | −0.7126 [hi] | 0.1715 [e] | 0.7847 [d] | 3.467 [a] |
| PP$_{5\%m}$ 8 g L$^{-1}$ | −0.5883 [gh] | −0.7225 [hi] | −0.7275 [hi] | −0.2375 [f] | 1.957 [c] |
| PP$_{5\%m}$ 16 g L$^{-1}$ | −0.9285 [i] | −0.4149 [fg] | −0.3875 [fg] | −0.7961 [hi] | 2.465 [b] |
| **LSD (0.05)** | | | 0.2834 | | |

Statistically significant differences are indicated by different letters (*p*-value ≤ 0.05, *n* = 4).

Comparable to the standard YEM, PP culture media exhibited a normal growth curve with no obvious lag period. Exponential growth with rapid increments in CFU counts was observed, reaching a plateau that remained for up to 10 days, before entering the decline phase. The PP culture media at a concentration of 16.0 g L$^{-1}$ supported rapid growth measured as doubling time (dt = 3.83 h) that was proportionate to the standard YEM (dt = 4.28 h). Doubling time was somewhat longer (dt = 6.33 h) in the case of 8.0 g L$^{-1}$ PPs (Figure 2b).

With respect to cell density measured as $OD_{600nm}$, the patterns of absorbance were similar across all of the tested culture media (Figure 2b). Cell density increased considerably with extending incubation time, being higher for YEM and followed by PP$_{5\%m}$ 16 g L$^{-1}$ and PP$_{5\%m}$ 8 g L$^{-1}$. ANOVA indicated significances attributed to the single effect of culture media (Table 2). However, growth levels increased with a longer time of incubation, and there were fewer differences among PP culture media. In particular, two-way interaction showed no significant difference between YEM and PP$_{5\%m}$ 16 g L$^{-1}$ at 14 days (Table 2).

The cell dry weight (DW) was positively correlated with the $OD_{600nm}$ and gradually increased during the incubation period. Noticeably, extended incubation favored greater DW that peaked at 14 days of incubation (~1.37–3.17 g L$^{-1}$, Figure 2b, Table 2). With longer incubation, the maximum DW production of PP$_{5\%m}$ 16 g L$^{-1}$ (2.60 g L$^{-1}$), not PP$_{5\%m}$ 8 g L$^{-1}$, was nearer to that of YEM (3.17 g L$^{-1}$), where no significant differences were reported by two-way ANOVA (Table 2).

Viable cell counts are a relevant measure for the assessment of culture media with potential for the production of agro-biopreparates. To make inferences on cell viability for the tested culture media, correlation analyses were performed among Log CFU mL$^{-1}$, $OD_{600nm}$, and DW. The result showed a strong positive correlation between DW and $OD_{600nm}$ levels (r = 0.84, *p* ≤ 0.0001), suggesting that the variation in these two parameters mainly reflected changes in total cell numbers of the tested rhizobia (Figure S3). As well, a significant correlation was detected between $OD_{600nm}$ and CFU counts

(r = 0.38, $p \leq 0.05$; Figure 2c), suggesting that viable cell counts partly explain the variance in total cell density. To further examine rhizobia cell viability in the tested culture media, estimates of nonviable cells (nVCs) were calculated, and a two-way ANOVA between culture media and incubation time was carried out (Figure 2b, Table 2). The nVCs decreased considerably during the first 24 h of incubation and were steadily maintained for five days. With longer incubation, nVCs increased variably among the tested culture media and were significantly highest for standard YEM followed by PP culture media, $PP_{5\%m}$ 8 g $L^{-1}$ and $PP_{5\%m}$ 16 g $L^{-1}$ (Table 2).

EC (mS cm$^{-1}$) and pH are used as indicators of metabolic activities of the tested rhizobia grown in various culture media. Following rhizobia inoculation into liquid batch cultures, the highest electric conductivity (1.94 mS cm$^{-1}$) was reported for $PP_{5\%m}$ 16 g $L^{-1}$ followed by $PP_{5\%m}$ 8 g $L^{-1}$ (1.22 mS cm$^{-1}$), and the lowest was observed for YEM (0.97 mS cm$^{-1}$). The coefficients of variation (CV%) for EC showed the lowest variation in $PP_{5\%m}$ 16 g $L^{-1}$ (2.80% ± 0.01) compared to $PP_{5\%m}$ 8 g $L^{-1}$ (3.60% ±0.01) and YEM (3.40% ± 0.00). Further analysis of EC fluctuations also indicated that the tested PP media had similar changes in EC (0.07–1.12) compared to YEM (0.08) (data not shown). In terms of pH, $PP_{5\%m}$ 16 g $L^{-1}$ exhibited the lowest CV (10.90% ± 0.10), followed by $PP_{5\%m}$ 8 g $L^{-1}$ (18.0% ± 0.20) and YEM (21.0% ± 0.28). Values of the average change in pH showed that the PP culture media exhibited the lowest fluctuations (0.0–0.02) in pH compared to YEM (0.13).

### 4.3. Fortified PP Preparations Improved Growth and Biomass Production of Tested Rhizobia

To improve the physical features and nutritional contents of plant pellets for biomass production, we tested the effect of increasing the molasses concentration of the processed pellets to 10%. To facilitate direct access and possible immobilization of rhizobia cells onto plant materials, plant pellets were included directly, not contained in teabags, to prepare culture media ($PP_{ntb10\%m}$ 8 g $L^{-1}$ and $PP_{ntb10\%m}$ 16 g $L^{-1}$). The efficiency of the developed culture media was compared to YEM and culture media based on teabags containing PPs prepared with 5% molasses. The cell density and biomass were statistically analyzed using ANOVA, based on independent and combined effects of culture media and incubation time (Table 3, Table S1).

**Table 3.** ANOVA of microbial biomass (dry weight (DW), g $L^{-1}$) of rhizobia grown in various combinations of culture media based on PPs within/without teabags compared to standard YEM. The least significant difference (LSD) for each analysis is shown.

| ANOVA: Single Effect (Culture Media) | | DW (g $L^{-1}$) | PPs Within Teabags | PPs Free, Without Teabags |
|---|---|---|---|---|
| | YEM | 3.29 [b] ± 0.35 | | |
| PPs Within Teabags | $PP_{5\%m}$ 8 g $L^{-1}$ | 1.35 [d] ± 0.14 | | |
| | $PP_{5\%m}$ 16 g $L^{-1}$ | 1.67 [d] ± 0.15 | | |
| | $PP_{10\%m}$ 8 g $L^{-1}$ | 1.24 [d] ± 0.07 | | |
| | $PP_{10\%m}$ 16 g $L^{-1}$ | 2.32 [c] ± 0.17 | | |
| PPs Free, Without Teabags | $PP_{ntb10\%m}$ 8 g $L^{-1}$ | 4.34 [a] ± 0.36 | | |
| | $PP_{ntb10\%m}$ 16 g $L^{-1}$ | 4.23 [a] ± 0.38 | | |

| ANOVA: Two-way Interaction (Culture Media and Time) | Culture Media | 4 h | 1 d | 5 d | 10 d | 15 d |
|---|---|---|---|---|---|---|
| | YEM | 2.231 [jklmn] | 2.749 [hijk] | 3.205 [fghi] | 4.059 [def] | 4.256 [cde] |
| PPs Within Teabags | $PP_{5\%m}$ 8 g $L^{-1}$ | 0.6473 [r] | 1.085 [qr] | 1.601 [mnopq] | 1.676 [mnopq] | 1.751 [lmnopq] |
| | $PP_{5\%m}$ 16 g $L^{-1}$ | 1.058 [qr] | 1.392 [nopqr] | 1.846 [lmnopq] | 1.9100 [klmnopq] | 2.147 [jklmno] |
| | $PP_{10\%m}$ 8 g $L^{-1}$ | 1.058 [qr] | 1.2560 [opqr] | 1.454 [mnopqr] | 1.318 [opqr] | 1.137 [pqr] |
| | $PP_{10\%m}$ 16 g $L^{-1}$ | 1.7057 [lmnopq] | 2.0198 [jklmnop] | 2.3340 [ijklm] | 2.6660 [hijk] | 2.9160 [ghij] |
| PPs Free, Without Teabags | $PP_{ntb10\%m}$ 8 g $L^{-1}$ | 2.6900 [hijk] | 3.761 [efg] | 4.833 [abcd] | 5.140 [abc] | 5.295 [a] |
| | $PP_{ntb10\%m}$ 16 g $L^{-1}$ | 2.591 [hijkl] | 3.454 [efgh] | 4.326 [bcde] | 5.223 [ab] | 5.600 [a] |
| LSD (0.05) | | | | 0.9120 | | |

Statistically significant differences are indicated by different letters ($p$-value $\leq 0.05$, $n = 3$).

Taking the dual effects of culture media and incubation time into consideration, generally all tested culture media supported progressive increases in cell density ($OD_{600nm}$), reaching a plateau on the tenth day of incubation (Figure 3a, Table S1). YEM supported significant increases in cell density compared to PPs packaged in teabags, 8 g $L^{-1}$, irrespective of the molasses concentration. The increase in teabag pellet concentration to 16 g $L^{-1}$ significantly increased cell density, comparable to YEM in the presence of 5% molasses and even higher with 10% concentrations of molasses. Teabag-free PP culture media recorded optimal cell density (2.30 with $PP_{ntb10\%m}$ 16 g $L^{-1}$ and 1.20 with $PP_{ntb10\%m}$ 8 g $L^{-1}$). One-way ANOVA identified YEM as having higher cell density than PPs contained in teabags, either 8 g $L^{-1}$ or 16 g $L^{-1}$ in presence of 5% or 10% molasses. The direct exposure of plant pellets, not being contained in teabags, overturned the results, where cell densities of teabag-free PP culture media significantly equalized the cell growth and density of standard YEM.

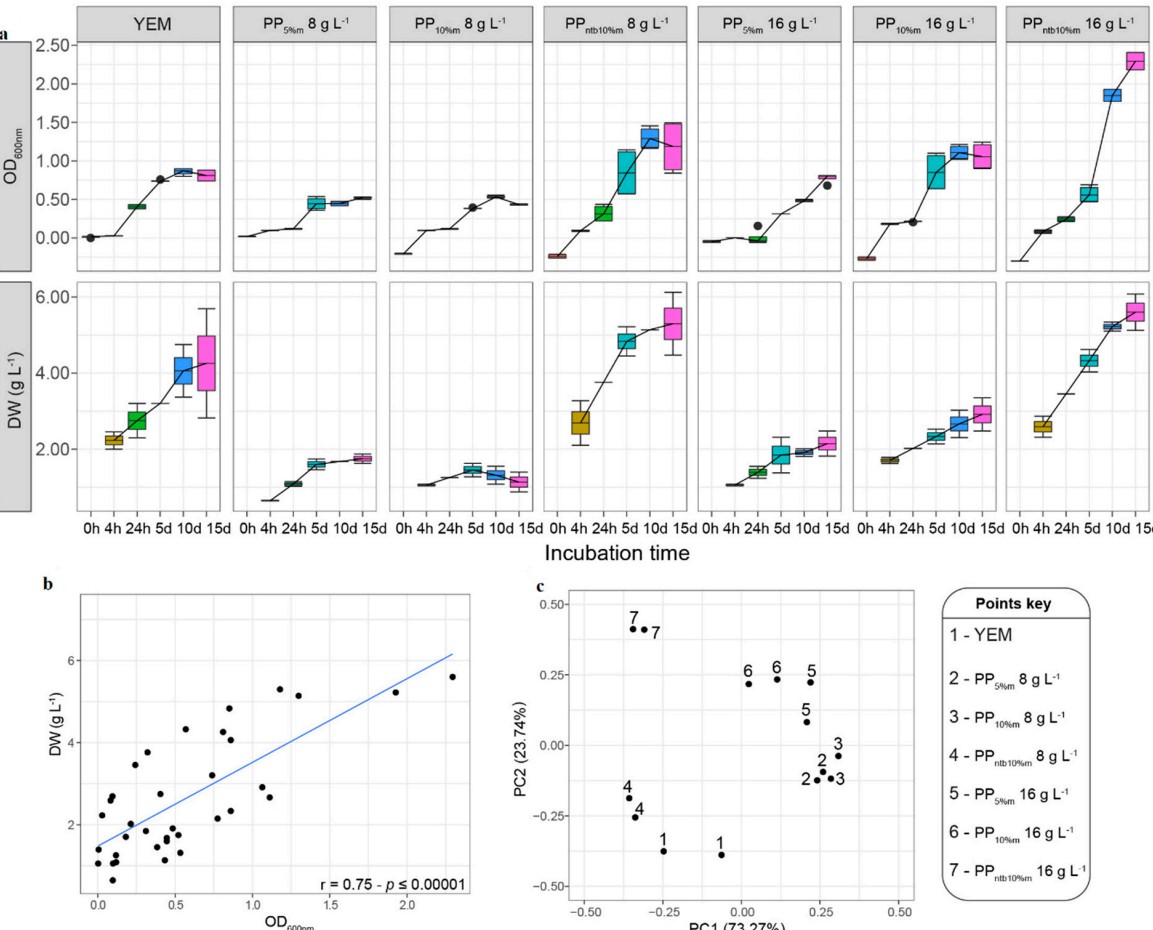

**Figure 3.** Biomass production of *R. leguminosarum* grown in batch cultures prepared from various combinations of culture media based on PPs within/without teabags compared to standard YEM. (**a**) boxplots of growth levels of rhizobia along the incubation period based on the tested parameters of optical density ($OD_{600nm}$) and dry weight (DW, g $L^{-1}$); (**b**) scatter plot of $OD_{600nm}$ and DW (g $L^{-1}$) values reported over the incubation period; (**c**) principal components analysis (PCA) generated from the recorded values of DW, $OD_{600nm}$, EC, and pH for each of the tested culture media. YEM, yeast extract mannitol; $PP_{5\%m}$ 8 g $L^{-1}$, culture media based on 8 g of plant pellets formulated with 5% molasses concentration (in teabags); $PP_{5\%m}$ 16 g $L^{-1}$, culture media based on 16 g of plant pellets formulated with 5% molasses concentration (in teabags); $PP_{10\%m}$ 8 g $L^{-1}$, culture media based on 8 g of plant pellets formulated with 10% molasses concentration (in teabags); $PP_{10\%m}$ 16 g $L^{-1}$, culture media based on 16 g of plant pellets formulated with 10% molasses concentration (in teabags); $PP_{ntb\ 10\%m}$ 8 g $L^{-1}$,

culture media based on 8 g of plant pellets formulated with 10% molasses concentration (teabag-free); $PP_{ntb\ 10\%m}$ 16 g $L^{-1}$, culture media based on 16 g of plant pellets formulated with 10% molasses concentration (teabag-free). All tested pellet formulations included 0.8% glycerol.

Dry weight values showed similar trends to $OD_{600nm}$, and the order of biomass production was concordant (Figure 3a, Table 3, Table S1). This was reflected through a positive correlation between the DW and $OD_{600nm}$ (r = 0.75, $p \leq 0.00001$; Figure 3b). ANOVA indicated a positive and significant single effect attributed to preparing culture media with PPs not contained in teabags (>4.2 g $L^{-1}$), compared to standard YEM (3.3 g $L^{-1}$). Two-way ANOVA showed that DW produced on all tested culture media increased by increasing incubation time, reaching climax in 10–15 days. The yield scored by teabag-free preparations surpassed (>5.20–5.60 g $L^{-1}$ dry weight) that of YEM (4.00–4.26 g $L^{-1}$) (Figure 3a, Table 3). This resulted in a gain of >25% in cell dry weight over standard YEM.

All PP culture media showed higher initial EC values than YEM that ranged from 1.20 to 2.68 mS $cm^{-1}$, and a final EC of 1.46–3.47 mS $cm^{-1}$. Such values coincided with relatively higher CVs ranging from 5.46% to 16.03%. Overall, PP culture media were significantly more conductive than YEM (0.99–1.04 mS $cm^{-1}$) and exhibited greater fluctuations (Table 4).

**Table 4.** One way ANOVA (culture media effect) of electric conductivity, pH, and corresponding coefficients of variation measured for batch cultures prepared from various combinations of culture media based on PPs within/without teabags compared to standard YEM.

| | **Culture Media** | | | | | | |
|---|---|---|---|---|---|---|---|
| **Variable** | **YEM** | **$PP_{5\%m}$ 8 g $L^{-1}$** | **$PP_{5\%m}$ 16 g $L^{-1}$** | **$PP_{10\%m}$ 8 g $L^{-1}$** | **$PP_{10\%m}$ 16 g $L^{-1}$** | **$PP_{ntb10\%m}$ 8 g $L^{-1}$** | **$PP_{ntb10\%m}$ 16 g $L^{-1}$** |
| $EC_0$ | 0.99 [e] * ± 0.00 | 1.70 [c] ± 0.00 | 2.28 [b] ± 0.02 | 1.40 [d] ± 0.00 | 2.45 [a,b] ± 0.00 | 1.20 [d,e] ± 0.00 | 2.68 [a] ± 0.00 |
| $EC_{min}$ | 0.98 [e] ± 0.00 | 1.70 [c] ± 0.00 | 2.10 [b] ± 0.00 | 1.40 [d] ± 0.00 | 2.45 [a,b] ± 0.00 | 1.07 [e] ± 0.01 | 2.68 [a] ± 0.00 |
| $EC_{fin}$ | 1.04 [e] ± 0.01 | 1.93 [c] ± 0.03 | 2.45 [b] ± 0.30 | 1.97 [c] ± 0.10 | 2.75 [b] ± 0.06 | 1.46 [d] ± 0.12 | 3.47 [a] ± 0.04 |
| $EC_{min}-EC_0$ | −0.01 | 0.00 | −0.18 | 0.00 | 0.00 | −0.13 | 0.00 |
| $EC_{fin}-EC_{min}$ | 0.06 | 0.23 | 0.35 | 0.57 | 0.30 | 0.39 | 0.80 |
| CV of EC | 3.39% | 5.46% | 10.79% | 16.03% | 5.55% | 12.67% | 11.50% |
| $pH_0$ | 7.28 [a] ± 0.02 | 7.20 [a] ± 0.10 | 7.25 [a] ± 0.05 | 7.00 [a,b] ± 0.00 | 6.50 [b] + 0.00 | 6.75 [b] ± 0.05 | 6.65 [b] ± 0.05 |
| $pH_{min}$ | 5.10 [d] ± 0.1 | 5.50 [c] ± 0.00 | 6.00 [b] ± 0.00 | 6.40 [a] ± 0.10 | 6.30 [a,b] ± 0.00 | 5.68 [c] ± 0.09 | 5.47 [c] ± 0.04 |
| $pH_{fin}$ | 5.63 [c] ± 0.05 | 6.50 [b] ± 0.00 | 6.75 [a,b] ± 0.05 | 6.85 [a] ± 0.05 | 6.83 [a] ± 0.02 | 6.36 [b] ± 0.06 | 6.64 [a,b] ± 0.02 |
| $pH_{min}-pH_0$ | −2.19 | −1.70 | −1.25 | −0.60 | −0.20 | −1.08 | −1.19 |
| $pH_{fin}-pH_{min}$ | 0.53 | 1.00 | 0.75 | 0.45 | 0.53 | 0.69 | 1.18 |
| CV of pH | 12.74% | 9.88% | 6.97% | 6.26% | 4.32% | 6.09% | 7.09% |

* Monitored initial (X0), minimum (Xmin), and final (Xfin) values of electric conductivity (EC) and pH of tested batch cultures of *R. leguminosarum* grown on YEM compared to formulated PP culture media with or without (noted with subscript "ntb") teabags. Coefficients of variation (CV%) changes (Xfin−X0) and maximal changes (Xmin−X0) are calculated for further comparisons. Statistically significant differences are indicated by different letters ($p$-value ≤ 0.05, $n$ = 3).

In general, increasing the PP concentrations to 16 g $L^{-1}$ and molasses to 10% had a positive effect on EC levels that led to greater fluctuations. Interestingly, the same molasses concentration corresponded to lower EC but increased variation across all tested PPs based on 8 g $L^{-1}$. Compared to YEM, all tested PP culture media showed the potential to maintain pH levels much closer to the initial values. Overall, the PPs exhibited lower CV in terms of pH (4.32%–9.88%) as well as lower maximal changes (−0.20 to −1.70) compared to YEM (−2.19), an effect that also coincided with the use of 10% molasses in PP teabag formulations.

Principal components analysis (PCA) was used to discern similarities among the standard YEM culture medium and PP culture media based on all tested parameters ($OD_{600nm}$, DW, EC, and pH). This produced a first major component that accounted for 73.27% of the total variation and mainly clustered all PP culture media based on 16.0 g $L^{-1}$ separate from YEM and all 8.0 g $L^{-1}$ PP culture media. The second major component accounted for 23.74% of the variation and pooled the teabag-free preparations together with YEM, separate from all teabag preparations (Figure 3c).

## 5. Discussion

Our previous research introduced and documented the challenge of plant-only-based culture media for increasing in vitro cultivability of rhizobacteria [32–35,37]. Here, the developed plant pellets represent a novel application of plant-only-based culture media potentiating the use of plant materials for value-added biomass production of rhizobacteria on a large scale. The pellets pool the nutrient stores of plant materials augmented with rich agro-byproducts into a formulation that is easy to handle, store, and deliver vegan nutrients for the convenient preparation of plant-based culture media.

Securing basic nutrient requirements for microbial cultivation is a fundamental step in the development of a culture medium capable of supporting biomass production of rhizobacteria [20,21,46]. Here, the composition of all PP formulations were sufficient to support good growth of *R. leguminosarum*. Critically, when used without teabags, PPs produced significantly greater dry weight than YEM. The richness of PP composition is supported by chemical analysis of the tested Egyptian clover powder, which was shown to have a wide C:N ratio of 12.20 and constitute carbohydrates, 42.00%; proteins, 21.60%; fiber, 22.80%; and ash, 13.30% [35]. In particular, the presence of high concentrations of aspartate (2.80 mg L$^{-1}$) and glutamate (~1.50 mg L$^{-1}$) amongst 17 other amino acids, as well as vitamins A and B2 in clover powder, is important for rhizobia physiology in planta [33,35]. These amino acids are involved in nitrogen-fixing symbiosis, ammonia uptake regulation, and facilitation of transamination reactions, as well as synthesis of glutamine, arginine, polyamines, peptidoglycans, and nucleic acids [35,47]. Further, the plant material is known to contain additional constituents of organic acids and secondary metabolites. Given that they are among the causal signals in root exudates that initiate plant–microbe interactions [6], their presence in the plant material may underpin the ability of PP culture media to cultivate and enhance the growth of rhizobia. For instance, flavonoids manipulate the growth and nod factors of rhizobia, stimulating symbiotic interactions that lead to nitrogen fixation [48–50]. However, since the presence and accumulation of these compounds in plants may be affected by abiotic and biotic factors, further chemical analysis of the plant materials and PPs are required to give insight on the compounds directly affecting the growth of rhizobia. More generally, this would also shed light on how the method of preparation of plant-only-based culture media promotes the growth of rhizobacteria.

The supplements of molasses and glycerol are of added value to the nutrient profile of PPs, likely enforcing an increased C:N ratio, favorable for biomass production. Increases in the C:N ratio plausibly explain why 16 g PP formulations with 10% molasses yielded greater biomass than formulations based on 5% molasses. That the tested molasses increases the C:N ratio is supported by previous studies reporting it to constitute 64% carbon and 6% nitrogen. In particular, its sugar content is constituted mainly of sucrose, 68%; glucose, 18%; and maltose, 13%. Molasses is also rich in minerals such as Fe (78 mg L$^{-1}$) and Mg (1370 mg L$^{-1}$). Since the supplements were used at their reported optimal concentrations [42,43,51] to develop PPs, the tested pellet combinations possessed a composite nutritional profile that is sufficient to feed rhizobia and to achieve the high demands of their inocula [52,53]. This assumption is consistent with metabolic analyses on the diverse pathways of rhizobia capable of utilizing a variety of carbon sources, with particular preference towards glycerol, sorbitol, and sucrose within *R. leguminosarum* [26,27]. It remains to be explored whether pellet combinations using other plant substrates (e.g., hay, maize, or alfalfa) with or without other agro-byproducts (such as whey, baker's yeast effluent, corn steep liquor) could also be used to cultivate rhizobia as well as other agro-biopreparates. To this end, plant powder fineness, substrate nutrients and ratios (plant material and/or additives), moisture content, and gelling potential should be considered for such approaches and the establishment of best plant pellet formulation practices for culture media preparation. Furthermore, targeting of genes related to rhizobia symbiosis, growth, or survival (e.g., *Nod*, *Rsh*, and *Rap*) will allow tangible assessments of the individual and combined actions of different pellet substrates [15,42].

To gauge the ability of PP culture media to support in vitro growth of *R. leguminosarum*, CFU counts were screened throughout extended incubation periods. The similarity of growth patterns in the PP culture media compared to YEM confirmed their ability to sufficiently support the growth of the tested

rhizobia, reaching $> 10^7$ mL$^{-1}$ culture. Moreover, growth vigor was highlighted through the calculated doubling times, particularly evident in the case of PP$_{5\%m}$ 16 g L$^{-1}$ and within the reported range of $\leq 6$ h reported for fast-growing rhizobia [54]. Of particular interest is that CFU count fluctuations of PP-based culture media in the exponential phase of growth appeared to mimic fed-batch modes of cultivation [55], highlighting the nutrient multiplicity of PPs and suggesting a step-wise fashion of substrate utilization of the tested rhizobia. Together with lower nVCs, these observations point towards unique features of PP-based culture media that support sustained growth and viability of rhizobia. Such features may be further investigated through stringent analysis of rhizobia growth in semi-continuous or continuous cultivation systems using PP culture media.

As the frontline output of agro-biopreparate application, the cell dry weight (DW) production potential of PP culture media was the main focus of this study. While the produced levels of cell DW were generally supported by observations of the optical density, their discordances may be attributed to heterologous protein production and/or pellet residues that interfere with gravimetric estimations of cell dry weight [54,55]. To this end, alternative biomass measurements based on the estimation of total $CO_2$ and/or online monitoring systems may improve the accuracy of cell biomass estimations [55,56].

Of interest is that increasing the molasses concentration from 5% to 10% led to significantly greater DW and OD$_{600nm}$ levels when 16.0 g L$^{-1}$ of PPs, not 8.0 g L$^{-1}$, was used. The effect could likely be attributed to the higher content of plant nutrients, not generally nitrogen or carbon content in the former compared to the latter preparation, but specifically growth factors and secondary metabolites required for rhizobia growth, e.g., multiple amino acids and vitamins [33]. This may be further studied through integrated approaches investigating the extent to which PPs mirror the chemical composition and concentration of the composite plant material and supplements, as well as how rhizobia respond to the changes in the stoichiometry of pellets components, an important factor that affects their metabolism and growth [12,53,57–59].

To further enhance pellet output and to facilitate PP application, we assessed the biomass potential of teabag-free preparations, i.e., direct inclusion of PPs into culture media. This further simplified and quickened the process of preparing plant-based culture media and recovered optimum biomass (Figure 3a, Table 3). Critical to industrial settings is that rhizobia did not appear to go through a lag phase on the onset cultivation, instead, in the case of PP$_{ntb10\%m}$ 16 g L$^{-1}$, yielding biomass at shorter time periods than YEM. Since teabag-free preparations also yielded the greatest biomass in terms of OD$_{600nm}$ and DW levels, PP residues have a positive effect on *R. leguminosarum* growth. One possible explanation is that, in addition to their rich and diverse nutrient stores, pellet residues in the culture media may act as surfaces available for immobilization, biofilm formation, and exopolysaccharide (EPS) production. The latter features enhance the root colonization potential, survival, and communication of rhizobia and are attributed to their sustainable presence in stressed environments whereby nitrogen fixation capabilities are hindered due to nutrient limitation, drought, salinity, and other abiotic stresses [60–62]. In this context, teabag-free preparations are recommended for mass cultivation and immobilization of agro-biopreparates. However, it remains to study the relationship between the physical and chemical properties of pellets and biofilm formation. In particular, pellet diameter, compactness, and density [63] may be factors affecting the available surface area for attachment of rhizobia cells to the pellet material.

With regards to the culture media store of electrolytes (EC), the correspondence between concentrations of PPs or molasses and the EC values suggests the presence of high concentrations of electrolytes within the plant material and the molasses supplement (Table 4). These results further indicate that the pellet–molasses combinations elicit multiple metabolic responses from the tested rhizobia. On the other hand, the lower pH fluctuations of the PPs that corresponded to increasing pellet and molasses concentrations highlight the desirable buffering capacity of PP culture media in agreement with considerations of culture media conductivity to promote metabolic functions [64]. These observations suggest that the tested PP culture media possess electrolytic properties favorable for promoting sustained biomass production and, possibly, for facilitating biofilm formation of

rhizobia [27,64]. PP culture media efficiency may be further assessed through rhizobia EPS production as well as electrokinetic potential, which are important in rhizobia symbiosis with host plants [65].

## 6. Conclusions

The present study encourages a new line of research focusing on the use of plant pellets as a vegan feedstock for value-added biomass production of rhizobacteria. Pellets and/or granulated materials are a common form of nutrient delivery to various living organisms including microorganisms. Therefore, we introduced pellets with the belief that they are convenient and easy to handle, store, and prepare plant-based culture media with, compared to other plant substances, e.g., powder and/or juices. Our results show that PPs cultivate rhizobia successfully and can yield greater biomass than the current standard YEM. Furthermore, several properties render the PPs as promising candidates for value-added commercialization of agro-biopreparates (e.g., bio-fertilizers and bio-pesticides): (1) PPs can integrate the rich nutritional stores of both plant materials and agro-byproducts, (2) PPs can be stored and simply added to distilled water for culture media preparation, (3) PPs can yield greater biomass at relatively short time periods, and (4) PPs exhibit favorable electrolytic properties for microbial cultivation. These properties also support PPs' use for plant microbiome cultivation and biomass production as well as microbiome delivery and transplantation towards good agricultural practices (GAPs) that are crucial for sustainable agriculture. Their physico-chemical properties, shelf-life, and application for growth and biomass production of other members of rhizobacteria remain to be tested. This newly developed plant pellet technology is currently under patent procedure.

**Supplementary Materials:** The following are available online at http://www.mdpi.com/2071-1050/12/20/8389/s1. Figure S1: Biomass yield of preliminary experiments, Figure S2: Colony and cell morphology (phase contrast 400× magnification) of *R. leguminosarum* cells grown in plant pellets (PPs) culture media (**panels a and b**) compared to standard YEM (**panels c and d**), Figure S3: Scatterplot of dry weight (DW, g L$^{-1}$) and optical density (OD$_{600nm}$) of *R. leguminosarum* grown on plant pellets culture media and the standard culture medium Yeast extract mannitol across the 14-day incubation period, Table S1: ANOVA analyses of microbial biomass (Optical density, OD$_{600nm}$)) of *R. leguminosarum* grown in various combinations of culture media based on PPs within/without teabags compared to standard YEM.

**Author Contributions:** Conceptualization, N.A.H., S.R., and H.-S.A.D.; Methodology, H.-S.A.D., M.A., H.H.Y. and M.A.H.; Software, M.A.H., M.S.S. and H.-S.A.D.; Validation, H.-S.A.D., M.A., M.A.H.; Formal Analysis, H.-S.A.D., M.A., M.A.H., N.A.H.; Investigation, H.-S.A.D., M.A., H.H.Y. and M.A.H., N.A.H.; Resources, N.A.H., S.R., M.T.A. and M.E.-T.; Data Curation, H.-S.A.D., M.A.; Writing—Original Draft Preparation, H.-S.A.D. and M.A. and N.A.H.; Writing—Review & Editing, H.-S.A.D., N.A.H., M.A.H., H.A.G., R.H., M.F., S.R.; Visualization; M.S.S., H.-S.A.D. and M.A.H.; Supervision, N.A.H.; Project Administration, N.A.H. and S.R.; Funding Acquisition, S.R. and N.A.H. All authors have read and agreed to the published version of the manuscript.

**Funding:** This research was funded by the German-Egyptian Research Fund (GERF-STDF 5032) and the APC was funded by IGZ, Germany.

**Acknowledgments:** Hegazi acknowledges the support of Alexander von Humboldt Stiftung, for equipment subsidies and financing his research stays at IGZ, Germany, and of the German Academic Exchange Service (DAAD) for funding Cairo University student training at IGZ, Germany and Cairo University, Egypt. We are also grateful to the technical laboratory skills and field support provided by fellow postgraduates Mohab Khalil and Saad Abdelwakeel.

**Conflicts of Interest:** The authors declare no conflict of interest.

## Abbreviations

Abbreviations for terminology used in this study.

| Term | Abbreviation |
| --- | --- |
| 1-aminocyclopropane-1-carboxylate | ACC deaminase |
| Analysis of variance | ANOVA |
| Bio-fertilizers and bio-pesticides | Agro-biopreparates |
| Byproducts of agro-industries | Agro-byproducts |
| Coefficients of variation | CV |
| Colony forming units | CFUs |

| Term | Abbreviation |
|---|---|
| Dry weight | DW |
| Electric conductivity | EC |
| Estimates of nonviable cells | nVCs |
| Exopolysaccharide | EPS |
| Good agricultural practices | GAPs |
| Optical density measured at 600 nanometers | $OD_{600nm}$ |
| Plant-growth-promoting rhizobacteria | PGPRs |
| Plant pellets | PPs |
| Potential of hydrogen | pH |
| Principal components analysis | PCA |
| Root zone | Rhizosphere |
| VC | Viable cells |
| Yeast extract mannitol | YEM |

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
