# Peer review of "Plant Pellets: A Compatible Vegan Feedstock for Preparation of Plant-Based Culture Media and Production of Value-Added Biomass of Rhizobia"

_sustainability, doi:10.3390/su12208389_

Round 1

Reviewer 1 Report

Dear authors,

The manuscript you presented shows a novel way to prepare medium to grow microorganisms.

The use of pellets is a novelty in this field, and opens the door to new ways of preparing those media.

You have some comments and corrections in the atached document. And I encorage you to change the begining of the abstract. The firts sentence is not apropiate for an abstract.

Sincerely,

Author Response

Dear Reviewer, 

We very much appreciate your comments. To improve our manuscript, we have responded to the comments point by point in the attached document. 

Sincerely, 
Hassan Daanaa

Reviewer 2 Report

Review of paper.

Presented article has been prepared in sufficient quality, many and various analysis and research results have been carried out and they were compared with the results of other researchers. Research results of presented paper are sufficient original and new, but the novelty of this paper is not sufficiently highlighted. The manuscript is partly structured in agreement with instructions to authors, and the quality of presented paper must be improved.  

Comments and remarks of reviewer:

  • The “Abstract” have only a general character, it provides only a little information on the results of the studies performed, and there are not presented some concrete research (analysis) data.
  • I propose that the authors provide a list of abbreviations at the beginning of the manuscript, as the authors use a wide variety of abbreviations that are difficult to remember when reading the text.
  • The article must be properly formatted: especially noticeable inaccuracies in the layout of the text and tables; uneven indentation of the text from the left edge (use of Tabs); in Table 2, it is not necessary to mark vertical and some horizontal lines; parts of the Tables 2 and 3 have been moved to the next page, they need to be placed on one page.
  • The authors need to explain more clearly the novelty of the research presented in the article.
  • The Discussion chapter is too wide, and the most important results of the research are not highlighted. This should be done in the Conclusions, which are not presented in the paper.

So, I can recommend a publication of this manuscript after a minor revision.

Author Response

Dear Reviewer,

We thank you for your comments and to improve the quality of the presented manuscript, we have responded accordingly in the attached document.

Sincerely, 
Hassan Daanaa
